# Evaluation of Cytotoxicity and Bioimaging of Nitrogen-Vacancy Nanodiamonds

**DOI:** 10.3390/nano12234196

**Published:** 2022-11-25

**Authors:** Claudia Fryer, Patricia Murray, Haifei Zhang

**Affiliations:** 1Department of Chemistry, University of Liverpool, Liverpool L69 7ZD, UK; 2Department of Molecular Physiology and Cell Signalling, University of Liverpool, Liverpool L69 3BX, UK

**Keywords:** nanodiamonds, fluorescent nanodiamonds, cytotoxicity, bioimaging, cell tracking

## Abstract

Nanodiamonds, due to their chemical inertness and biocompatibility, have found extensive uses in drug delivery and biomedical applications. Fluorescent nanodiamonds with fluorescent properties generated by nitrogen-vacancy defects have been intensively investigated for bioimaging, due to their high quantum yield and high photobleaching stability. In addition, the surface properties and particle size of nanodiamonds have significant impacts on cellular uptake and imaging quality. In this study, nitrogen-vacancy nanodiamonds with different particle sizes (40 nm and 90 nm) have been physicochemically characterised and investigated for their cytotoxicity and potential in fluorescence imaging. The nanodiamonds (with concentrations up to 100 µg/mL) showed cell viability >70% with mesenchymal stromal cells. The number of nanodiamonds was observed to have a larger impact on cell viability than the mass of nanodiamonds. Larger nanodiamonds (90 nm) exhibited a lower level of cytotoxicity, higher cellular uptake and fluorescence intensity. The results indicate the potential of using fluorescent nanodiamonds as a nanoprobe for effective bioimaging and cell tracking.

## 1. Introduction

Nanodiamonds (NDs) first emerged in the 1960s, but have attracted more attention in the last few decades [1]. The rigid structure of diamonds is made up of a dense network of sp^3^ hybridised carbon atoms, each with tetrahedral symmetry. Each carbon atom forms four bonds with neighbouring carbon atoms, resulting in a lack of free valence electrons that contributes to the high inertness of nanodiamonds. However, as for other nanoscale materials, the properties of NDs are predominantly due to their surface chemistry [2]. For NDs, their high surface area gives rise to properties that are different to the bulk material because surface carbon atoms cannot form four bonds; hence, terminal groups, such as hydrogen and hydroxyl groups, are present to reduce unfavourable dangling bonds.

Both the intrinsic structure (sp^3^ carbon network) and the surface chemistry of NDs can be exploited to alter material properties, such as stability, optical and electronic properties. The tunability aspect, combined with their high biocompatibility and large-scale synthesis capability, has meant that NDs are being explored for many potential biomedical applications [3,4,5].

Diamonds can be prepared by chemical vapour deposition (CVD), high-pressure high-temperature (HPHT) and detonation methods [2]. CVD, as the name suggests, involves the deposition of chemical vapour onto a substrate, resulting in the formation of diamond films. Nanocrystalline and ultra-nanocrystalline films with grain sizes of 5–100 and 3–5 nm, respectively, can be prepared [6]. NDs are more commonly prepared by HPHT or detonation of carbon explosives [7]. HPHT conditions are required to form NDs from graphite, in order to overcome the high energy barrier [8]. The process generally involves irradiation and annealing, as well as milling techniques to achieve smaller particle sizes. The high-energy process results in defects within the diamond structure where the carbon atoms are knocked out, resulting in vacancies. When these vacancies are near to nitrogen centres, produced by the addition of nitrogen compounds in the starting material, nitrogen-vacancy (NV) defects are formed [2,9]. This results in unique absorbance/fluorescence properties, which can be exploited for bioimaging. It is now possible to prepare small particle sizes and doped bright fluorescent nanodiamonds using the HPHT process [10,11,12,13]. The detonation method can also be employed, where carbon explosives are detonated by the generation of a shock wave, which causes high temperatures and pressures to initiate explosion; this results in conditions of 3000–4000 °C and 20–30 GPa within the chamber [2].

When a nitrogen atom is near to a vacancy within the ND structure, it is known as a NV centre or defect. Some NV defects are intrinsic to the high-energy process utilised to synthesise NDs; however, they can also be intentionally introduced to produce fluorescent NDs. NV defects are caused by irradiating the NDs with high-energy particles, such as electrons, protons or helium ions, followed by thermal annealing under vacuum [14,15].

Most examples of NDs with NV defects are synthesised by HPHT, although it is also possible to incorporate nitrogen in the detonation process. NV defects have been found in detonation NDs after electron irradiation and annealing; however, intense and stable emission is usually observed in the larger NDs (>30 nm) [16]. Detonation NDs have many structural defects, both intrinsic and surface-associated, which may result in fewer NV centres. It has been confirmed that NV defects can exist in smaller detonation NDs. However, only a relatively small amount can exist due to the small particle size; hence, there is limited detectable fluorescence [17].

There are two types of NV defects that have different fluorescence properties, the neutral NV^0^ and negative NV^−^ defects [18]. The NV^−^ defect tends to be the more dominant product of the irradiation process, which results in excitation and emission wavelengths of ~560 nm and 700 nm, respectively, and a quantum efficiency of ~1 [19,20]. These wavelengths are particularly useful for imaging, as there is limited cell autofluorescence in the far-red NIR region. The high biocompatibility of NDs, combined with the fluorescence properties of the NV^−^ defect (e.g., long fluorescence lifetime, high quantum yield and low photobleaching), is particularly attractive in bioimaging [8,18,19].

NDs with NV^−^ defects were used to investigate sentinel lymph node mapping in a mouse model and a fluorescent signal could still be detected 37 days post-injection (subcutaneous or intraperitoneal) [21]. This highlights the potential of their long-term imaging capability. In addition, 200 nm fluorescent NDs were functionalised with poly(glycerol) and mannose for increased retention in sentinel lymph nodes. The in vivo visualization of the lymph nodes demonstrated the potential for cancer diagnostics and guided surgery [22]. Milled NDs, with a mean particle size of 46 nm, have also shown good uptake in HeLa cells, where the uptake mechanism was most likely clathrin-mediated endocytosis [23]. It was found that NDs predominantly resided in endocytic vesicles; however, smaller NDs appeared to be free in the cytoplasm. The multimodal imaging potential of NDs in cancer cells was successfully demonstrated by fluorescence imaging, atomic force microscopy, and 3D soft X-ray tomography [24].

Despite the advances in NDs for bioimaging, studies on the physicochemical properties of NDs, bioimaging capabilities and the correlation between them are still limited and are required. In this study, NV NDs (40 nm and 90 nm) have been characterised and assessed for their biocompatibility and their potential as nanoprobes for fluorescence imaging. Their effectiveness as cell-labelling nanoprobes for fluorescence imaging was investigated by assessing their ability to label mesenchymal stromal cells (MSCs). MSCs were used because this cell type has been extensively used as regenerative medicine therapy in animal models, as well as in clinical trials [25,26]. The size effect of the NV NDs was investigated and correlated with the cytotoxicity and cell uptake results obtained by flow cytometry and confocal laser microscopy.

## 2. Materials and Methods

### 2.1. Materials

All reagents were used as received from Merck Life Sciences (Gillingham, UK). NV NDs (40 nm and 90 nm) were provided by Element Six Ltd. (Didcot, UK).

### 2.2. Characterisation Methods

A Malvern Zetasizer Nano instrument and the associated Zetasizer software were used to determine the hydrodynamic diameter, polydispersity index and zeta potential of particles in water via dynamic light scattering (DLS)( Malvern, UK). Samples were diluted with water (0.1 mg/mL) and sonicated for 10 min prior to DLS analysis. Images of the particles were acquired with a Hitachi S4800 scanning electron microscope (SEM) at 5 kV. For SEM imaging, the nanodiamond suspension was processed by adding a drop of the nanosuspension onto an SEM (Hitachi Europe Ltd, Berkshire, UK) stub and allowing the solvent (water) to evaporate. Solid samples were directly attached to the stub using conductive double-sided carbon tape. Infrared spectroscopic data were obtained using a Vertex 70 Fourier Transform Infrared (FTIR) Spectrometer (Bruker, Brighton, UK). Optical properties were investigated using a μ-Quant Microplate Reader and a Fluorescence Lifetime Spectrometer (FLS) 1000 (Edinburgh Instruments). To obtain fluorescence spectra, samples were prepared at a concentration of 0.1 mg/ml and analysed using a front-face sample holder, 590 nm emission filter and excitation/emission bandwidths of 0.8–3 nm. Excitation wavelengths of 500 or 560 nm were used and emission was measured in the range of 550–800 nm, with a 1 nm step and 0.2 s dwell time.

### 2.3. Cell Culture

The murine mesenchymal stromal cell line D1 (ATCC) was used for all in vitro cell studies. MSCs had previously been modified to express firefly luciferase, under the control of the constitutive promoter, EF1α [27]. MSCs were cultured in 6 cm or 10 cm tissue culture dishes (Greiner CELLSTAR^®^, Sigma Aldrich, Gillingham, UK) in high-glucose Dulbecco’s Modified Eagle Medium (DMEM, Sigma Aldrich, Gillingham, UK), containing 10% fetal bovine serum (FBS, Sigma Aldrich), 1% non-essential amino acids (Sigma Aldrich) and 2 mM L-glutamine (Gibco). Cells were incubated at 37 °C and 5% *v*/*v* CO_2_ and passaged with 1% trypsin/EDTA (Sigma Aldrich) when they reached 90% confluence.

### 2.4. In Vitro Cell Studies

**Cell Viability.** CellTiter-Glo^®^ Luminescent Cell Viability Assay (Promega, Southampton, UK) was used according to the manufacturer’s guidelines. MSCs were seeded in triplicate at a density of 2 × 10^5^ cells/well in a 96-well plate and left to attach and grow for 24 h. After this, the medium was replaced with 200 µL of fresh medium, containing various concentrations of NDs (0–400 µg/mL). Cells were incubated for a further 24 h before carrying out the assay. Before performing the luminescence-based viability assay, the cells were washed to remove any excess NDs, thus reducing possible interference. Luminescence measurements were recorded with a µ–Quant Microplate reader.

**Flow Cytometry.** Flow cytometry was used to assess the cellular uptake and fluorescence properties of the ND-labelled cells. MSCs were seeded at a density of 2 × 10^6^ cells/well (24-well plate) and incubated for 24 h. After this, the medium was aspirated and replaced with fresh medium, containing nanoparticles of varying concentrations (0–100 µg/ml). After a further 24 h, cells were resuspended in PBS (1×) and kept on ice ahead of analysis. Flow cytometry was carried out using a BD FACSCalibur™ (BD Biosciences, Wokingam, UK) with a 488 nm laser wavelength and a long pass FL3 filter (670 nm). When appropriate, threshold values for side scatter (SSC), front scatter (FSC) and FL3 were altered relative to the control to exclude unwanted events. In addition, 10,000 events were counted and the data were analysed with Flowing Software 2.5.1 (Turku Bioscience Centre, Turku, Finland). A sample that was not labelled with any of the NV-NDs was measured and served as a control for “background fluorescence”. Levels of fluorescence above this background level in the labelled samples were considered to be due to the NV-NDs.

**Confocal Laser Microscopy.** Confocal laser microscopy was employed to visualise and determine the location of NDs in MSCs. MSCs were seeded in 8-well chamber slides (Ibidi µ-Slide 8-well) at a density of 2 × 10^5^ cells/well. After 24 h of labelling with nanoparticles (50 or 100 µg/mL), the samples were fixed with 4% PFA (paraformaldehyde, Sigma Aldrich). Cells were permeated with Triton-X (0.1% *v*/*v*) and stained with 4′,6-diamidino-2-phenylindole (DAPI, 1:1000, Thermo Fisher Scientific, Warrington, UK) and Phalloidin-AlexaFluor 488^®^ (1:50, Invitrogen, Warrington, UK). Images were acquired with a Zeiss LSM 800 Airyscan confocal microscope using 405 nm, 488 nm and 640 nm diode lasers and a 63× oil objective. Z-stack images were obtained using a step size of 0.34 μm.

## 3. Results

NV NDs are nanoparticles that carry some intrinsic absorbance/fluorescence properties, resulting from the high-energy synthesis of NDs [1,2,9,14]. This fluorescence property, along with other physicochemical properties, means that the NV NDs are a highly promising cell-labelling probe for bioimaging [8,21,22,23,24]. We investigate here the characterisation of NDs and their size impact on cytotoxicity, uptake by MSCs, and fluorescence imaging.

### 3.1. Characterisation of NV NDs

Two NV ND samples were evaluated, with estimated particle sizes of 40 nm (NV-40) and 90 nm (NV-90) based on the manufacturer’s information. Table 1 summarises the DLS measurements where the particle size was similar to the expected size (Figure 1) and a low polydispersity index was measured (~0.1), suggesting a narrow particle size distribution. Both NV NDs had a negative zeta potential, but NV-40 was more negative. The negative zeta potential of the particles is expected due to the hydroxyl terminating groups from the synthesis of NDs [28]. The difference in zeta potential between the NV NDs can be explained by considering the particle sizes; zeta potential is calculated from electrophoretic mobility, which is size-dependent.

The absorption properties were investigated with UV–vis spectroscopy. The light scattering effects of the NV NDs were apparent, resulting in high background absorbance. Figure 2 shows the absorption profile of the NV NDs, where greater light scattering is observed for the larger NV-90 NDs, but no obvious absorbance peak is observed.

The fluorescence properties of the NV NDs are shown in Figure 3. The light scattering properties of the NDs proved to be an issue, so a filter was added to block emission below 590 nm. This enabled the detection of the peak associated with NV^−^ at ~700 nm, which was more pronounced for NV-90 NDs (Figure 3b); these results agree with similar experiments in the literature [29]. As a control, NDs without NV defects (HPHT NDs 100 nm) were also analysed using the same excitation wavelengths and only a background signal was observed.

### 3.2. In Vitro Evaluation of NV NDs as Fluorescent Probes

#### 3.2.1. Cytotoxicity

The CellTiter-Glo^®^ assay was carried out to investigate whether NV NDs had an effect on the viability of MSCs. For the NV NDs, cell viability decreased with increasing concentration and NV-40 NDs appeared to have a greater cytotoxic effect than NV-90 NDs, as shown in Figure 4. Cell viability remained >70% for both sets of NV NDs at all dosing concentrations.

In order to ascertain whether the greater cytotoxic effect of NV-40 was due to particle size or number, the assay was repeated and cells were dosed by the number of NV NDs rather than mass concentrations. The number of NDs was calculated using the average density of commercial NDs (3.5 g cm^−1^) [30,31]. Figure 5 highlights that NV NDs had a similar effect on cell viability when similar particle numbers were used, hence suggesting that the increased cytotoxicity of NV-40 observed previously was due to the cells being exposed to a greater number of particles. Therefore, it is likely that more particles were taken up by the MSCs to cause the reduced viability.

#### 3.2.2. Assessing Cellular Uptake with Flow Cytometry

Flow cytometry was used to assess the uptake of NV NDs into MSCs. For both types of NV NDs, there was an increase in fluorescence intensity with increasing ND concentrations; however, this was less pronounced with NV-40 NDs, as shown in Figure 6.

The difference between NV-40 and NV-90 was more obvious when comparing the percentage of cells positive with NDs (Figure 7). The percentage of labelled cells increased with increasing concentrations for both sets of ND; however, following labelling with NV-90 NDs, a greater proportion of the MSC population became fluorescent. Above 50 µg/mL, >80% of the cell population were labelled with NV-90 NDs.

Following the cytotoxicity results, which highlighted the differences when MSCs were dosed based on mass concentration and estimated number of NDs, a flow cytometry experiment was carried out on MSCs that had been dosed with various estimated numbers of NV NDs. In this case, there was very little change in fluorescence intensity between the untreated cells and those dosed with various numbers of NV-40 NDs, as shown in Figure 8. However, for NV-90 ND, fluorescence intensity increased with increasing number of NDs.

When comparing the percentage of cells that were positive, the uptake of NV-90 NDs increased with increasing numbers of NDs (Figure 9). At the highest number (1.8 × 10^10^), >40% of cells had taken up NV-90, which is consistent with the results for the equivalent mass-based dosing concentration of ~10 µg/ml. The flow cytometry data suggest that NV-90 NDs show more intense fluorescence than NV-40 NDs. Higher numbers of NV-40 NDs were required to observe increases in fluorescence intensity and even then, it was not as high as for NV-90.

#### 3.2.3. Visualising Cellular Uptake with Confocal Microscopy

Labelled cells were fixed and stained with DAPI and AlexaFluor^®^ 488 phalloidin to show the nucleus and actin cytoskeleton, respectively. NV-90 NDs were much more abundant and brighter compared to NV-40 NDs when imaged at the same laser intensity (Figure 10). The more effective visualisation of the NV-90 ND is likely to be attributed to their more intense fluorescence, which is consistent with the results of the flow cytometry analysis (Figure 6).

## 4. Discussion

NDs with NV defects (NV-40, 40 nm; NV-90, 90 nm) were evaluated for cytotoxicity with MSCs and their potential for cell tracking and bioimaging. Their particle size was confirmed by DLS and the low polydispersity index (~0.1) for both samples indicated a relatively narrow particle size distribution. The zeta potential of both NDs was negative; however, NV-40 was more negative, which can be explained by the fact that the measurement was derived from electrophoretic mobility, which is size-dependent. In addition, a long-standing issue with standardising ND synthesis exists and so many of the properties of NV NDs vary. This is due to the large number of factors that influence its defects, such as particle size, the percentage of graphitic form of carbon, surface functionality, ND density and refractive index (RI) of the layers [18]. Hence, the difference in zeta potential could be related to the number of NV^−^ centres near the surface.

The absorption properties of the NV NDs were investigated with UV–vis spectroscopy, as displayed in Figure 2. The light scattering properties were evident, which is consistent with data reported in the literature and can be explained by the knowledge that NDs have a high RI of 2.42 [2,32]. NV NDs usually show absorbance at ~560 nm, which corresponds to the emission at ~700 nm, a characteristic of the major defect (NV^−^) that can be observed in Figure 2 [19].

NV NDs were evaluated in in vitro cell studies to assess cytotoxicity (ATP assay), labelling efficiency (flow cytometry) and intracellular localisation (confocal microscopy). It was found that cell viability decreased with increasing concentrations and it appeared that NV-40 NDs were more cytotoxic. This was investigated further by dosing MSCs by the ND number rather than mass concentration, which was estimated based on an approximate density of 3.5 g cm^−3^ for commercial NDs and assuming perfect spherical morphology [30,31]. In this case, it was found that NV-40 and NV-90 NDs had a similar effect on cell viability, suggesting that the amount of NDs, rather than the particle size, caused the cytotoxic effect (Figure 4). Much higher cell viability was observed in this experiment, compared with mass concentration dosing, as the numbers of NDs used did not reach the higher dosing concentrations used previously.

Flow cytometry showed greater increases in fluorescence intensity and a higher percentage of fluorescent cells when MSCs were dosed with NV-90 NDs (Figure 6). At the highest concentration of 100 µg/mL, 88% and 57% of the cell population were labelled with NV-90 and NV-40 NDs, respectively. When comparing MSCs dosed with different numbers of NDs, rather than mass concentration, MSCs dosed with NV-40 NDs displayed negligible fluorescence (Figure 8). When cells were labelled with 1.8 × 10^10^ NV-90 NDs, >40% cell uptake was observed. The flow cytometry results from dosing by ND numbers suggested that NV-90 NDs had stronger fluorescence.

The NV NDs were subsequently visualised in MSCs using confocal microscopy. NV-90 NDs could be observed in the majority of cells, whereas it was more difficult to observe NV-40 NDs (Figure 10). There are two possible reasons for this difference; either fewer NV-40 NDs are taken up by cells or they emit lower levels of fluorescence. The latter scenario is more likely because for confocal imaging, MSCs were dosed at 100 µg/mL, so there should be more NV-40 NDs than NV-90 NDs. In addition, cellular uptake was observed for all NDs in this study using flow cytometry. Therefore, it is likely that NV-40 NDs emit lower levels of fluorescence than NV-90 NDs. This statement agrees with the literature, as larger NDs tend to have more NV defects [17]. The staining pattern observed in this study suggests that the NV-NDs are localised in the endolysosomal compartment, which is consistent with reports showing that NDs are taken up by endocytosis [33]. Although we only explored the fate of the NV-NDs and their effect on the MSCs over a 24 h period, a previous study has demonstrated that following uptake in human MSCs, NDs showed no effect on cell viability and were retained within lysosomes over a 10-day period (the experimental end point), without any evidence of exocytosis [34]. This suggests that NDs would be suitable for longer-term cell tracking, with the main limitation likely being due to the number of NDs per cell being halved at each cell division, rather than as a result of the NDs being lost from the cells via exocytosis.

## 5. Conclusions

Fluorescent NV NDs (NV-40, 40 nm and NV-90, 90 nm) have been characterised and evaluated for cell cytotoxicity (ATP assay) and as probes for fluorescence imaging (assessed by flow cytometry and confocal laser scanning microscopy). The results indicate that the number of NDs has a greater impact on cell viability, compared to the mass of NDs taken up by the cells. NV-40 NDs were found to be more toxic to the MSCs and also displayed lower fluorescence intensity in the cells. In contrast, NV-90 NDs were found to be superior to the smaller NV-40 NDs in terms of cytotoxicity, cellular uptake, fluorescence intensity, and fluorescence imaging quality. The high fluorescence of the NV-90 NDs, in combination with their capacity for high dosing due to their inherent biocompatibility, indicate that NV-90 NDs are promising candidates for biomedical applications and are a particularly useful tool for determining the fate and biodistribution of administered cells using fluorescent imaging techniques.

## Figures and Tables

**Figure 1 nanomaterials-12-04196-f001:**
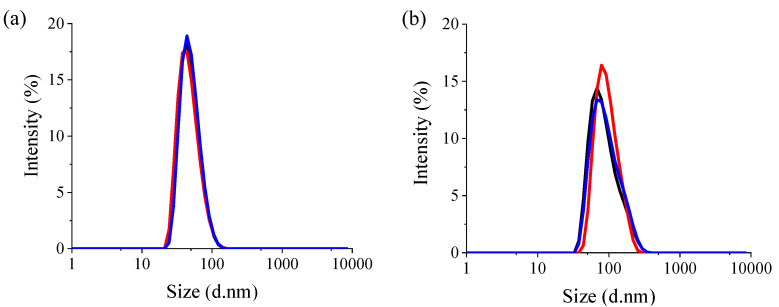
Volume particle size distribution of (**a**) NV-40 and (**b**) NV-90 NDs (0.1 mg/mL). The different colours represent 3 measurement scans with DLS.

**Figure 2 nanomaterials-12-04196-f002:**
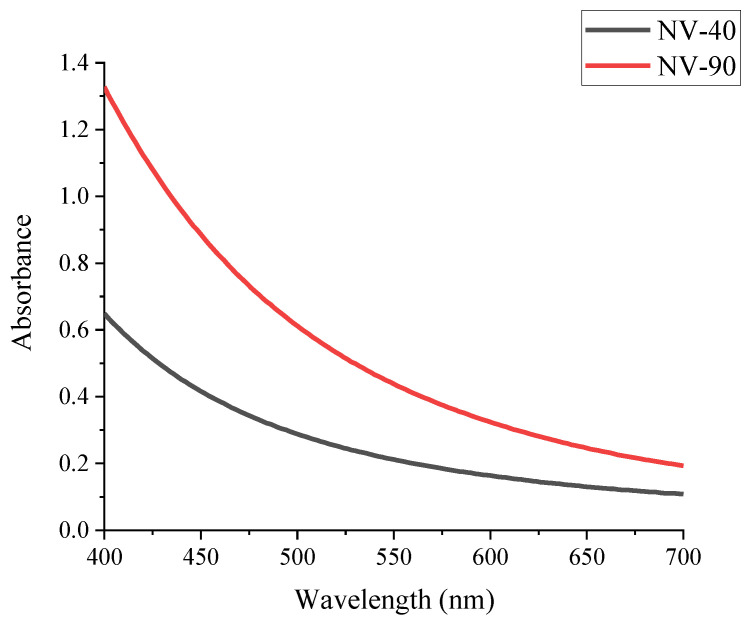
Visible spectra of NV NDs (water, 0.1 mg/mL).

**Figure 3 nanomaterials-12-04196-f003:**
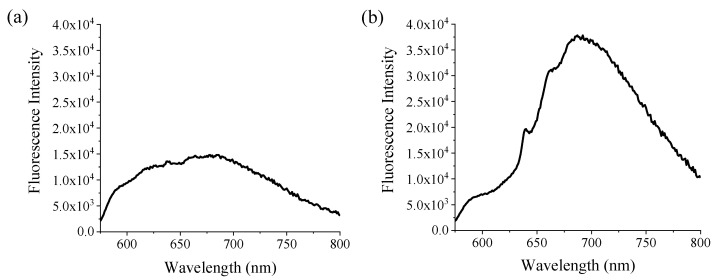
Emission spectra (λ_ex_ = 560 nm) for (**a**) NV-40 and (**b**) NV-90 NDs (0.1 mg/mL water). An emission filter was used to block emission below 590 nm to counteract the effects of light scattering.

**Figure 4 nanomaterials-12-04196-f004:**
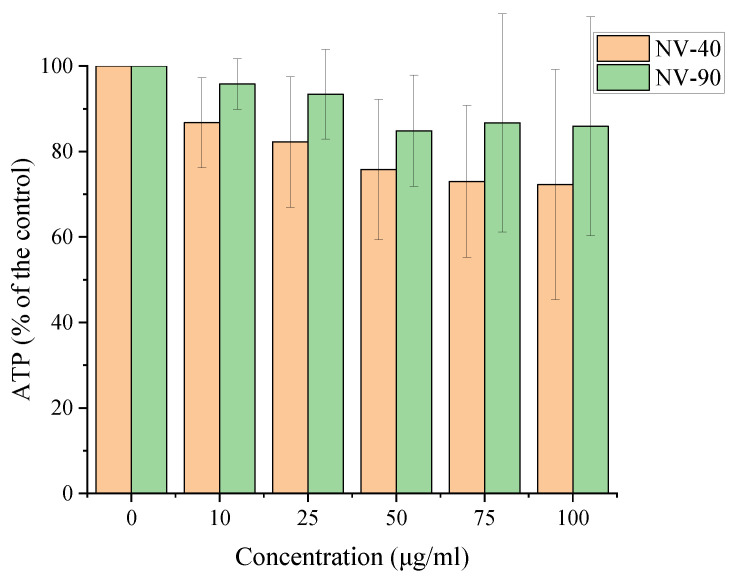
The effect of NV NDs on ATP production in MSCs. Cells were seeded at a density of 2 × 10^5^ cells/well for 24 h and NDs were then incubated with MSCs for a further 24 h ahead of measuring ATP.

**Figure 5 nanomaterials-12-04196-f005:**
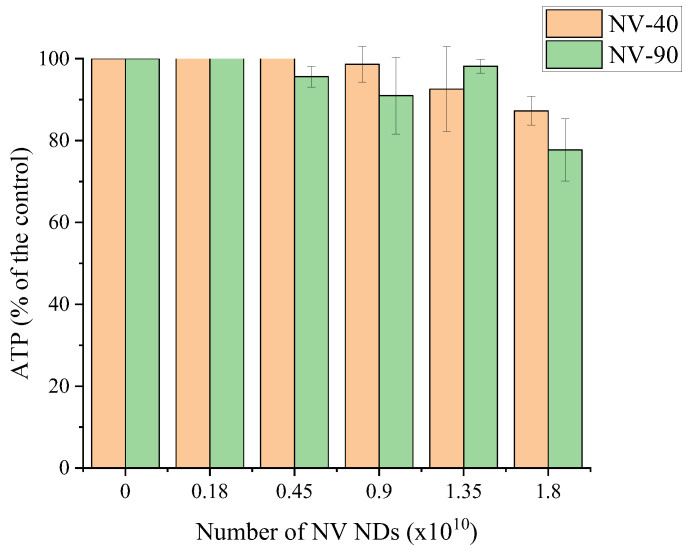
The effect of NV NDs on ATP production in MSCs. Cells were seeded at a density of 2 × 10^5^ cells/well for 24 h and NDs were then incubated with MSCs for a further 24 h ahead of measuring ATP.

**Figure 6 nanomaterials-12-04196-f006:**
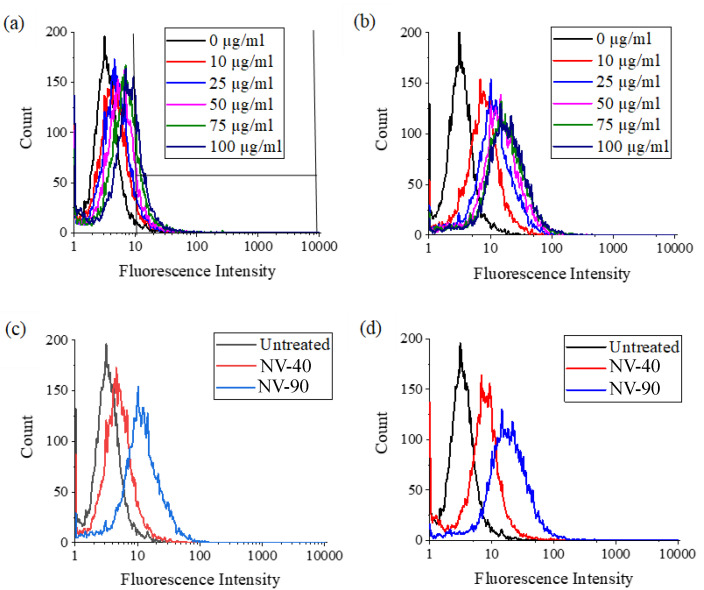
Live cell flow cytometry of MSCs in PBS labelled with various concentrations of (**a**) NV-40 and (**b**) NV-90 NDs for 24 h. The fluorescence intensity of MSCs labelled with both NV NDs at concentrations of (**c**) 25 µg/ml and (**d**) 100 µg/mL is also shown. Fluorescence intensity was measured using the FL3 filter (670 LP), where 10,000 events were counted, and the fluorescence gating used for all experiments was the same as in (**a**).

**Figure 7 nanomaterials-12-04196-f007:**
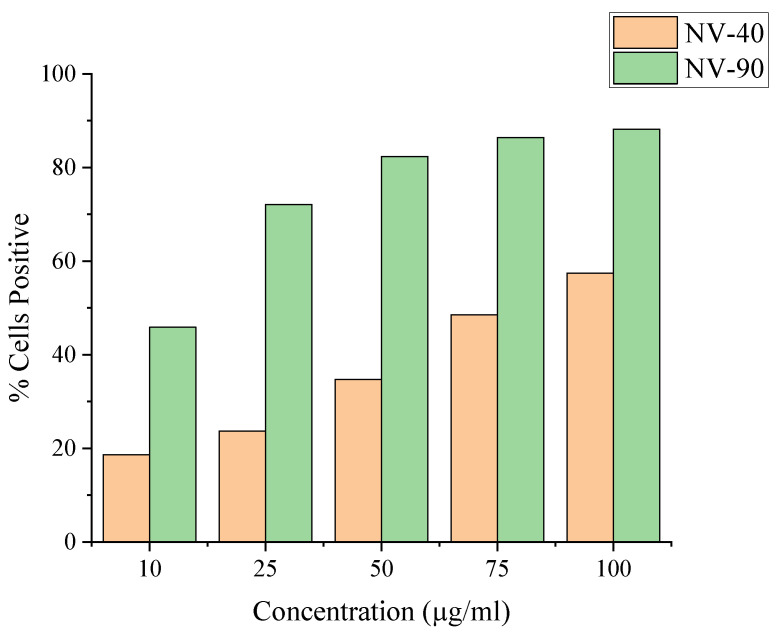
Comparison of the percentage of cells positive with various concentrations of NV-40 and NV-90 NDs. Fluorescence intensity was measured using the FL3 filter (670 LP), where 10,000 events were counted.

**Figure 8 nanomaterials-12-04196-f008:**
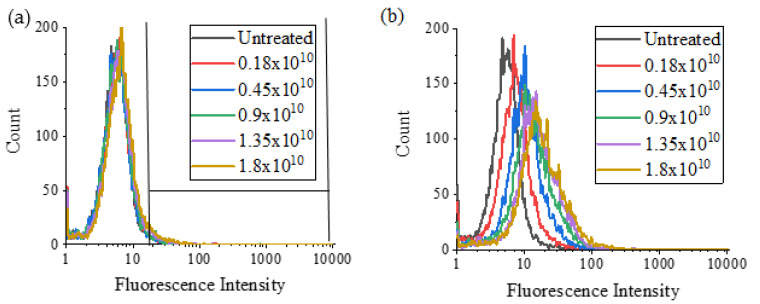
Live cell flow cytometry of MSCs in PBS labelled with various numbers of (**a**) NV-40 and (**b**) NV-90 NDs for 24 h. Fluorescence intensity was measured using the FL3 filter (670 LP), where 10,000 events were counted, and the fluorescence gating used for all experiments was the same as in (**a**).

**Figure 9 nanomaterials-12-04196-f009:**
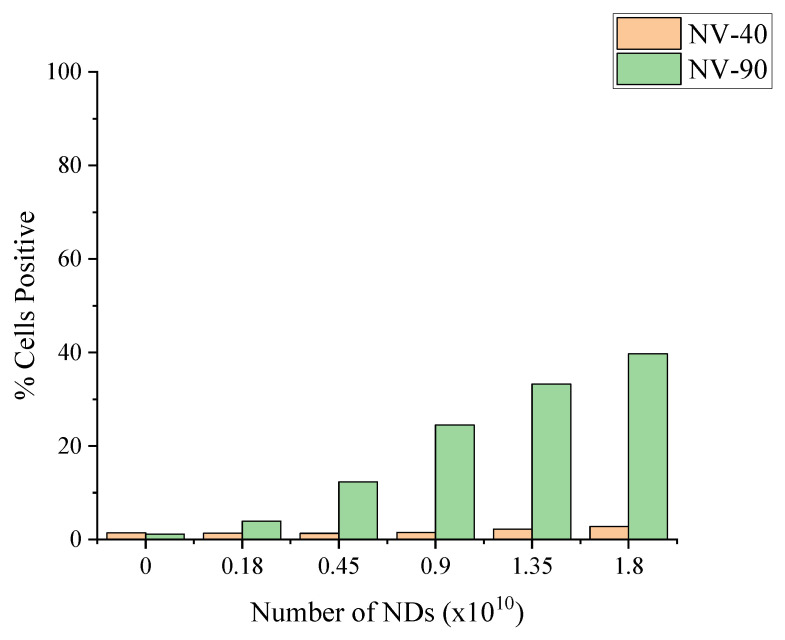
Comparison of the percentage of cells positive with various numbers of NV-40 and NV-90 NDs. Fluorescence intensity was measured using the FL3 filter (670 LP), where 10,000 events were counted.

**Figure 10 nanomaterials-12-04196-f010:**
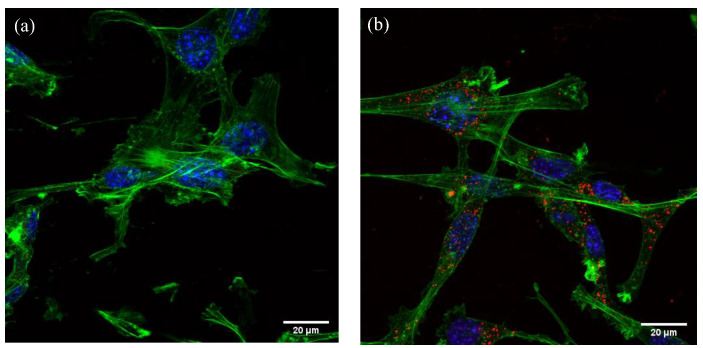
Confocal microscope images of MSCs dosed with (**a**) NV-40 and (**b**) NV-90 NDs. MSCs were seeded in chamber slides (2 × 10^5^ cells/well) and left to grow and adhere for 24 h, before dosing with 100 µg/mL NDs for a further 24 h. Cells were fixed with 4% PFA and stained with DAPI (blue) and AlexaFluor^®^ 488 (green); NV NDs are displayed in red.

**Table 1 nanomaterials-12-04196-t001:** DLS analysis of NV NDs.

ND Sample	Expected Particle Size (nm)	Intensity Particle Size Distribution (nm)	Polydispersity Index	Zeta Potential (mV)
NV-40	40	49 ± 17	0.11	−46 ± 1
NV-90	90	93 ± 46	0.13	−35 ± 1

## Data Availability

The data presented in this study are available upon request from the corresponding author.

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
