# Peer review of "Evaluation of Cytotoxicity and Bioimaging of Nitrogen-Vacancy Nanodiamonds"

_nanomaterials, 2022, doi:10.3390/nano12234196_

Round 1
Reviewer 2 Report
The presented manuscript clearly describes the effect of two types of fluorescent ND on the viability of a selected cell line and their potential for fluorescent imaging. The useful point is that concentration vs number of apllied particles is compared to deduce which is the main factor for the observed in vitro effects.
I have few recommendations to improve the presentation:
1. The information in Fig.1 and Table 1 is redundant and one of them can be moved to SupplMaterials
2. The cytotoxicity/vialbility assay is based on luminenscence measurements that partly overlap with the region of intrinsic absorbance and luminenscence of the studied NDs. The authors should provide more information on these effects, e.g. fluorescence spectra of the NDs solutions (as applied to the cells) with the ATP assay luminenscence measurements parameters should be given in the Suppl.MAterials.
3. The selection of this particular type of cell line should be motivated.
4. More information should be added, either in the Introduction or in the Discusiion, on the fate of the NV-NDs inside the cells, beacuse the viability data are tested for 24 hrs only. This is short enough for the imaging purposes, but should be checked or commented whether a long-term vialbility is preserved. This question is provoked by the claimed long-term in vitro fluorescence (37 days) for some NDs.
In conclusion, I can recommend the manuscript for publication after considering the suggested corrections/clarifiactions, as it addresses a useful practical issues on the biomedical application of the studied fluorescent NDs.
Reviewer 3 Report
This is a good manuscript. It can be published after a revision. I’d like the revised version address the following issues.
The Introduction section should be abridged approximately twice. It is not a review of the literature. The authors should state the problem which will be solved, and the manuscript should be devoted to a solution of that problem.
The Figures 6a.b and 7b are mess. The authors should find a way for a readable presentation.
Line 177. Should be only “Visible” or “Vis”.
Figures 4,5 should be omitted and replaced by 1-2 sentences of comments.
Figure 1. Intensity measured in percentile is strange.
Figure 3. What is “Fluorescence Intensity”? Units?
One needs to take into account a background fluorescence (lines 153-155) when an intensity of fluorescence initiated by a light source is measured. I’d like to see a brief comment on this subject.
I would write in more reserved terms about the potential applications of ND-NV in the Conclusion section.
